# No Significant Association Between Vitamin C Supplements and Frailty in Korean Older Adults: A Cross-Sectional Analysis of the 2018–2019 Korea National Health and Nutrition Examination Survey

**DOI:** 10.3390/nu17182977

**Published:** 2025-09-17

**Authors:** Seung Guk Park, Hyoeun Kim

**Affiliations:** Department of Family Medicine, Inje University Haeundae Paik Hospital, Busan 48108, Republic of Korea; sgpark@paik.ac.kr

**Keywords:** frailty, older adults, vitamin C, dietary supplements, Korean

## Abstract

**Background/Objectives**: The association between vitamin C intake and frailty among older adults remains unclear, and evidence from Asian populations is limited. Using nationally representative data, we aimed to examine whether vitamin C supplementation is associated with frailty in Koreans aged ≥65 years. **Methods**: We analysed 2819 participants from the 2018–2019 Korea National Health and Nutrition Examination Survey. Participants were categorised as non-users of dietary supplements (*n* = 1517), users of other supplements (*n* = 1227), and vitamin C-only users (*n* = 75). Frailty was defined using a modified Fried phenotype comprising five components (weight loss, exhaustion, weakness, slowness, and low physical activity); individuals having ≥3 components were classified as frail. Multivariable logistic regression was used to estimate odds ratios (ORs) and 95% confidence intervals (CIs) for frailty by supplement use and total daily vitamin C intake from foods and supplements, adjusting for sociodemographic, lifestyle, and health factors. **Results**: Compared with non-users (adjusted OR: 0.65, 95% CI: 0.26–1.66) or users of supplements other than vitamin C supplements (adjusted OR: 0.81, 95% CI: 0.26–2.53), vitamin C supplementation was not significantly associated with frailty prevalence. Higher total intake showed a tendency toward lower frailty prevalence in crude analyses, but this was not significant after adjustment (*p* for trend = 0.120). **Conclusions**: In this nationally representative study of Korean older adults, vitamin C supplementation was not significantly associated with frailty. These findings contribute to the conflicting evidence on micronutrients and frailty and suggest that broader dietary quality, rather than single-nutrient supplementation, may be more important for healthy ageing.

## 1. Introduction

By 2025, Korea is expected to comprise a super-aged society, in which more than 20% of the total population will be over 65 years of age [1,2]. Healthy ageing is becoming more important as the life expectancy and geriatric population increase [3]. Although the incidence of frailty generally increases with age, frailty is not considered a healthy and normal part of ageing. Frailty is a medical condition in which physical functions deteriorate severely enough to interfere with daily living [2,4,5,6]. In 2001, Fried et al. presented the criteria for defining frailty which satisfied three or more of the following five items: weight loss, exhaustion, weakness, slowness, and inactivity [4]. Frailty is clinically relevant not only because it reduces quality of life and independence but also because it increases risks of morbidity, disability, and mortality in older adults [2,3,4,7].

Nutritional and physical interventions are therefore considered key to preventing or delaying frailty. Ageing is often accompanied by physiological changes, such as reduced taste and smell, dry mouth, and decreased gastric acid production, that affect the gastrointestinal system. Moreover, reduced food intake due to masticatory disorders, appetite loss, dysphagia, or constipation may further impair absorption of macro- and micronutrients, leading to nutritional imbalance and vitamin deficiency [8]. Although frailty is multifactorial, Fried et al. hypothesized that inadequate dietary intake could contribute to its development and that sufficient nutrient intake is important for healthy ageing [4]. Consistent with this, a Korean study found that frail older adults consumed significantly less calcium, protein, and vitamins A, B1, B2, E, and C compared with robust or pre-frail peers [9]. Similarly, a meta-analysis reported that reduced intakes of vitamins B6, C, E, and folate were associated with a higher risk of frailty [10]. The Invecchiare (ageing) in Chianti (InCHIANTI) study also explained that low intake of vitamins B9, C, D, E and protein increases the risk of frailty [11].

Inflammation and oxidative stress have also been implicated in frailty pathogenesis. In a large cohort study, Saum et al. demonstrated that markers of oxidative stress were associated with higher frailty risk [12]. Vitamin C is an effective antioxidant [12,13,14], may therefore play an important role. However, a Korean study showed that vitamin C intake from food alone was often insufficient, with 80% of older adults consuming less than the recommended nutrient intake (RNI) of 100 mg/day [15]. Given the widespread undernutrition of vitamin C in this population, inadequate intake may increase long-term frailty risk.

However, evidence on vitamin C and frailty remains inconsistent [5,16,17,18,19,20,21], and data from Asian populations are particularly limited. Given this gap, and the high prevalence of both frailty and vitamin C insufficiency in Korean older adults, this study aimed to investigate whether vitamin C supplementation is associated with frailty, and whether associations differ by intake amount, using nationally representative data from the 2018–2019 Korea National Health and Nutrition Examination Survey (KNHANES).

## 2. Materials and Methods

### 2.1. Data Source and Study Population

This study analysed data from the Korean National Health and Nutrition Examination Survey (KNHANES) conducted in 2018–2019. The KNHANES provides nationally representative health and nutritional information data of Koreans. It consists of a personal health interview and a health and nutrition examination survey. Our study analysed sociodemographic data from a health interview survey, body measurement and hand grip strength data from a health examination survey, food intake through the 24 h recall method, and dietary supplement (DS) data from a nutrition survey.

We selected individuals aged 65 years or older who participated in the KNHANES 2018–2019. Of the 16,109 participants, 3395 were 65 years or older. Among the 3395 participants, 315 who had missing values in the variable to determine current DS intake and 140 who had missing values in the corresponding variables (frailty component, sociodemographic, and nutritional intake-related variables) were excluded. In addition, 18 individuals with extreme values, such as total metabolic equivalent of the task (MET) for 1 week > 10,000 kcal, were excluded. Using this process, 2922 participants were selected. Of these, 1517 participants did not take any dietary supplements (non-DS), while 1405 reported the use of DSs. Among supplement users, 1227 took supplements other than vitamin C (other DS), and 178 were vitamin C users. The vitamin C user group was further subdivided into those who concurrently used other supplements in addition to vitamin C (*n* = 103) and those who exclusively used vitamin C supplements (Vitamin C, *n* = 75). For the final analysis, we compared three groups: non-DS (*n* = 1517), other DS (*n* = 1227), and Vitamin C (*n* = 75) (Figure 1). The reporting of this study followed the STROBE (Strengthening the Reporting of Observational Studies in Epidemiology) guideline for cross-sectional studies, and a completed checklist is provided in Appendix A. The KNHANES 2018–2019 data were approved by the Institutional Review Board of the Korea Center for Disease Control and Prevention (IRB Approval Number: 2018-01-03-P-A, 2018-01-03-C-A).

### 2.2. Definition of Frailty

We used a modified definition of the frailty phenotype developed by Fried et al. [4], adapted to the variables available in the KNHANES. This operationalisation has been previously applied in several KNHANES-based studies [7,9,22]. Frailty was assessed across five domains: weight loss, exhaustion, weakness, slowness, and low physical activity (Table 1). Several adaptations were necessary due to the survey design. Specifically, exhaustion was assessed using perceived stress because CES-D fatigue items were not included in KNHANES; slowness was evaluated based on EQ-5D mobility items instead of measured walking speed. These modified criteria have demonstrated associations with health indicators including nutritional status, pulmonary function, and oral health, thereby supporting their construct validity in this population.

Participants were classified as frail if they met ≥3 criteria, pre-frail having 1–2 criteria, and robust having none of the five criteria. This study divided frailty into non-frail (Robust and Pre-frail) and frail (Frail). In supplementary analyses, each frailty component was modelled as a separate dependent variable (Appendix A), and the results were generally consistent. We acknowledge that these operational definitions may introduce misclassification; however, no consistent pattern across components was observed, suggesting that differential misclassification by exposure is unlikely. In addition, prior Korean studies have demonstrated the construct validity of these proxies and reported consistent findings.

### 2.3. Definition of Vitamin C Supplements

The DS data obtained through nutritional surveys were classified into 12 types: multivitamins and mineral supplements, vitamins A, C, and D, omega-3 fatty acids, probiotics, red ginseng, calcium, lutein, propolis, iron, and others. To evaluate the association between vitamin C supplementation and frailty, we defined the vitamin C group as participants who used vitamin C supplements exclusively, thereby excluding individuals who concurrently used other supplements. Furthermore, because individuals who are in better health are generally more likely to take DSs, participants using supplements other than vitamin C were included as an additional comparator group to reduce potential selection bias. Information on DS use, including vitamin C, was collected via self-reported questionnaires. Because supplement use was assessed by self-report rather than objective verification (e.g., pill counts, prescription records, or biomarkers), potential recall bias and misclassification cannot be ruled out.

In secondary analyses, frailty status was additionally evaluated according to total daily vitamin C intake from both dietary sources and supplements. While the primary exposure of interest in this study was the use of vitamin C supplements, this exploratory analysis was performed to investigate a potential dose–response relationship between overall vitamin C intake and frailty. To further assess the relationship between the amount of vitamin C intake and frailty, total vitamin C intake (from food or food plus DSs) was categorised into four groups: <100 mg (below the recommended daily amount in Korea), 100–999 mg (reflecting the upper limit [UL] of 1000 mg suggested by the WHO), 1000–1999 mg, and ≥2000 mg (the suggested UL of daily vitamin C intake in Korea) [25].

### 2.4. Sociodemographic Factors

Sociodemographic factors included age, sex, education, household income level, living arrangements, smoking and drinking status, body mass index (BMI), and medical history of chronic diseases or cancer. The total population aged 65 years or older was subdivided into those aged 65 to 74 years and those over 75 years. Educational level was divided into below 9 years of school (below middle school education) or over 9 years of school (high school education or above). Household income was divided into the lowest quartile (Q1) and other income levels (Q2, Q3, and Q4). Living arrangement was defined as living together or alone (unmarried, divorced, or separated).

Smoking status was divided into non-smoker, former smoker, or current smoker. Drinking habit was categorised into drinking once a week or less and frequent drinking more than two times a week. Based on BMI, which was calculated by dividing weight (kg) by the square of height (m^2^), participants were classified as underweight (BMI < 18.5 kg/m^2^), normal weight (18.5 ≤ BMI < 23 kg/m^2^), overweight (23 ≤ BMI < 25 kg/m^2^), and obese (BMI ≥ 25 kg/m^2^) [26]. Chronic conditions were categorised as either three or more, or two or fewer, of the following: hypertension, stroke, myocardial infarction, angina pectoris, diabetes, chronic kidney disease, and cancer, such as gastric, hepatocellular, colon, breast, cervical, lung, thyroid, and others.

### 2.5. Statistical Analyses

The KNHANES employs a multistage, stratified, cluster sampling design to generate nationally representative estimates of the Korean population. Accordingly, all analyses incorporated stratification variables (strata: kstrata), cluster variables (primary sampling unit, PSU), and sampling weights. Continuous variables were compared across groups using survey-weighted general linear models, and categorical variables using χ^2^ tests with complex sample design correction. Daily vitamin C intake from food and supplements showed a skewed distribution and was therefore log-transformed to approximate normality. Quartile cut-off points derived from the transformed values were used to classify participants into four intake categories (<100, 100–999, 1000–1999, and ≥2000 mg/day), which were then applied in the regression analyses.

Survey-weighted logistic regression models were fitted to estimate crude and multivariable-adjusted odds ratios (ORs) with 95% confidence intervals (CIs) for the association between vitamin C supplement use and frailty. Multivariable models were adjusted for age, sex, education, income, living arrangements, body mass index, smoking status, drinking frequency, presence of chronic disease, total energy intake, and protein intake. Physical activity, being a component of the frailty phenotype, was not additionally included as a covariate to avoid construct overlap and potential over-adjustment. Adjusted predicted probabilities of frailty for each group were calculated from fitted models using post-estimation predictive margins. Absolute risk differences (ARDs) with 95% CIs were derived from these margins, and pairwise group comparisons were conducted using post-estimation contrast procedures. All analyses were performed using Stata version 15.0 SE (Stata Corp, College Station, TX, USA). A two-tailed *p* value < 0.05 was considered statistically significant.

## 3. Results

### 3.1. Baseline Characteristics

A total of 2819 participants aged ≥65 years were included in the analysis (non-DS: *n* = 1517; other DS: *n* = 1227; vitamin C only: *n* = 75, unweighted count, representing a nationally weighted population). Baseline characteristics are shown in Table 2. Compared with non-DS, vitamin C users had a lower proportion with ≤9 years of education (53.8% vs. 74.4%; *p* = 0.0015). Compared with other DS, vitamin C users had a lower proportion of women (40.5% vs. 60.6%; *p* = 0.0017) and lower fat intake (25.95 vs. 30.00 g/day; *p* = 0.0470). Total vitamin C intake from food and supplements was markedly higher in the vitamin C group than in either comparator (both *p* < 0.0001). Other baseline characteristics, including age, income, living status, smoking, alcohol intake, BMI, comorbidity burden, vitamin C intake from food, total energy and protein intake, did not differ significantly in pairwise comparisons. The proportions of frail participants were 17.3% in non-DS, 12.6% in other DS, and 9.0% in the vitamin C group, while robust participants comprised 16.6%, 20.8%, and 30.4%, respectively. Robust status was more frequent in vitamin C users compared with non-DS (*p* = 0.0109), whereas no significant difference was observed between vitamin C and other DS.

### 3.2. Association Between Vitamin C Supplementation and Frailty

Results of survey-weighted logistic regression are shown in Table 3. Compared with non-DS, the adjusted OR for frailty in vitamin C users was 0.65 (95% CI, 0.26–1.66). Compared with other DS, the adjusted OR was 0.81 (95% CI, 0.26–2.53). Adjusted predicted probabilities were 16.2% (non-DS), 13.6% (other DS), and 11.6% (vitamin C). The corresponding absolute risk differences were −4.6% (non-DS vs. vitamin C; *p* = 0.308) and −2.1% (other DS vs. vitamin C; *p* = 0.656).

### 3.3. Frailty According to Total Vitamin C Intake

Frailty status according to daily vitamin C intake is shown in Table 4. As expected, intake from food only was uniformly low across all participants (mean < 100 mg/day), whereas supplement users had substantially higher intakes. This confirms that dose-related differences largely reflect supplemental intake rather than dietary sources. The prevalence of frailty was 16.8% in participants consuming < 100 mg/day, 9.9% in those consuming 100–999 mg/day, 10.3% in those consuming 1000–1999 mg/day, and 6.4% in those consuming ≥ 2000 mg/day (*p* for trend < 0.001). In adjusted models, the ORs for frailty were 0.79 (95% CI, 0.57–1.08), 0.79 (95% CI, 0.31–2.02), and 0.46 (95% CI, 0.05–3.68), respectively, compared with <100 mg/day. The overall trend did not reach significance (*p* = 0.120).

### 3.4. Frailty Components

As shown in Appendix A, vitamin C users had lower adjusted probabilities of weight loss (11.6% vs. 16.2%), exhaustion (8.6% vs. 15.8%), and slowness (15.4% vs. 25.2%) compared with non-DS participants, although these differences did not reach statistical significance. Weakness could not be estimated in the vitamin C group due to sparse data.

## 4. Discussion

In light of inconsistent findings from previous studies, our analysis of nationally representative Korean adults aged ≥65 years found no significant association between vitamin C supplementation and frailty prevalence. Although crude prevalence of frailty was lower among vitamin C users compared with non-users (9.0% vs. 17.3%), this difference was not statistically significant after multivariable adjustment.

Our findings are consistent with several previous studies that reported no significant associations between vitamin C and frailty. In the Framingham Offspring Study (*n* ≈ 2384, 11-year follow-up), adherence to a Mediterranean-style diet and higher carotenoid intake were associated with lower frailty risk, whereas vitamins C and E intake showed no associations [19]. The Colaus cohort in Switzerland (*n* ≈ 3300, 5.2 year follow-up) similarly found no association between vitamin C or E supplementation and changes in grip strength, a key frailty component [20].

By contrast, other studies have reported positive associations. The InCHIANTI study (*n* ≈ 1000, ≥65 years, 3-year follow-up) found that low intake of vitamins C, D, E, and folate predicted higher frailty risk [11]. A systematic review further highlighted that insufficient intake of antioxidant vitamins, including vitamin C, may contribute to frailty development [27].

These discrepancies across studies likely reflect heterogeneity in design (cross-sectional vs. longitudinal), baseline nutritional adequacy, frailty definitions, and exposure assessment methods (dietary recall vs. plasma concentrations), as well as differences in covariate adjustment. Some prior studies showing protective associations may have been affected by reverse causation, since frailer individuals might have been more motivated to take supplements. This could partly explain inconsistencies across cohorts. In our population, non-users consumed on average ~52 mg/day of vitamin C, below the Korean RNI of 100 mg/day, whereas supplement users consumed much higher amounts (mean ~1447 mg/day). This striking contrast reflects the Korean supplement market, where standalone vitamin C products are most commonly sold as 1000 mg tablets—the conventional form taken by consumers. Because most participants clustered at either very low or very high intake levels, we could not evaluate subtle dose–response relationships around the RNI threshold. However, if vitamin C exerted a strong linear protective effect, such an extreme contrast should have yielded clearer differences in frailty risk. The absence of significant associations even across this polarized intake gradient suggests that vitamin C supplementation alone is unlikely to substantially influence frailty risk in this population.

Biological plausibility remains. Vitamin C plays essential roles in collagen and carnitine synthesis and in antioxidant defence, all of which are relevant to muscle mass and physical function [14]. Deficiency induces muscle atrophy in animal models [28], and low vitamin C intake has been associated with reduced grip strength in older Korean women [29]. Because weakness and exhaustion are central components of frailty, vitamin C could theoretically influence frailty pathways. Nonetheless, given the multifactorial nature of frailty, supplementation with a single nutrient is unlikely to substantially modify risk.

This study has several strengths. We used a large, nationally representative dataset with standardised protocols and directly evaluated vitamin C supplementation in an Asian population where both frailty and vitamin C insufficiency are common. To minimise selection and healthy-user bias, we classified participants into three comparator groups—non-users, other DS users, and vitamin C—only users. By separating vitamin C—only users from other DS users without vitamin C (other-DS), who share similar underlying health-seeking characteristics but differ in vitamin C intake, we were able to better isolate potential vitamin C—specific associations while reducing confounding from general supplement-related healthy-user behaviours.

Several limitations should also be acknowledged. First, the cross-sectional design precludes causal inference. Second, supplement use and dietary intake were self-reported, introducing potential recall error, though standardized interviews and cross-checking with 24 h recalls improved reliability. Third, dietary intake was assessed using a single 24 h recall, which may not capture habitual consumption and could bias associations toward the null. Fourth, the use of modified frailty proxies (EQ-5D mobility for slowness, perceived stress for exhaustion, AWGS thresholds for weakness) may limit comparability with cohorts using direct measures, even though validated in prior Korean datasets. Fifth, the number of vitamin C–only users were relatively small in unweighted counts (*n* = 75), which reduced statistical power and widened confidence intervals. However, because KNHANES applies complex sampling weights, this subgroup still represents a much larger segment of the Korean older population, partially mitigating concerns about generalizability. Finally, although we adjusted for major sociodemographic, lifestyle, and nutritional factors (including protein intake), residual confounding cannot be ruled out. In particular, other dietary components, micronutrient intakes, medications, and the use of additional dietary supplements were not fully captured and may represent a potential source of residual confounding in this analysis. Moreover, supplement users—including both vitamin C–only and other DS groups—may have generally healthier behaviours and socioeconomic profiles, which we sought to reduce by structuring the analysis into three comparator groups.

Despite these limitations, our findings provide meaningful evidence. They indicate that while many older Koreans fail to achieve the RNI of 100 mg/day through diet alone, vitamin C supplementation was not significantly associated with reduced frailty prevalence. These results add to the heterogeneous international literature and suggest that vitamin C supplementation alone is unlikely to be a major determinant of frailty risk. From a public health perspective, improving overall dietary quality and ensuring adequate intake of multiple nutrients may be more effective strategies for preventing frailty than relying on single-nutrient supplementation.

## 5. Conclusions

This study found no significant association between vitamin C supplement use and frailty in a nationally representative sample of Korean older adults. While the results do not support vitamin C supplementation as an independent strategy for frailty prevention, they provide important evidence addressing inconsistent findings in the literature. Adequate nutrition and multifactorial strategies remain essential for promoting healthy ageing, with further longitudinal and interventional studies needed to clarify the role of vitamin C in this context.

## Figures and Tables

**Figure 1 nutrients-17-02977-f001:**
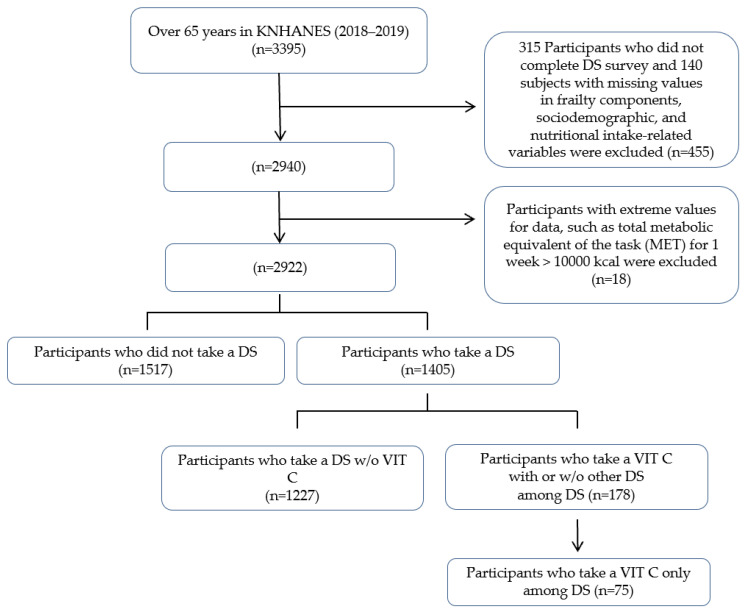
Flowchart of the participant selection and classification according to vitamin C supplement use in KNHANES 2018−2019. Abbreviation: KNHANES, Korean National Health and Nutrition Examination Survey; DS, dietary supplements; *n*, unweighted sample size; w/o, without; VIT C, vitamin C supplements.

**Table 1 nutrients-17-02977-t001:** A slightly modified definition of the frailty index using the KNHANES dataset.

Frailty Components	Indicator (Proxy in KNHANES)	Measurement/Cutoff (Scored as 1)	Reference(s)
Weight loss	Self-reported unintentional weight loss	Self-reported unintentional weight loss ≥ 3 kg in the past year	[7,9,22]
Exhaustion	Self-reported level of perceived stress	Response of “extreme stress” in daily life (KNHANES stress questionnaire)	[9]
Weakness	Hand grip strength	Digital dynamometer (T.K.K. 5401, Japan); maximum of three trials from dominant hand (six trials if ambidextrous); Asian working group for sarcopenia (AWGS) 2019 cutoffs: <28 kg (men), <18 kg (women)	[7,9,22,23]
Slowness	EQ-5D; Mobility limitation	I have some problems in walking, or I am confined to bed.	[7,9,22]
Low energy expenditure	The Global Physical Activity Questionnaire	The lowest 20% of physical activity evaluated by METs for each gender based on data from the National Survey of Older Koreans (METs: men < 494.65 kcal/week and women < 283.50 kcal/week)	[22,24]

Abbreviation: KNHANES, Korean National Health and Nutrition Examination Survey; MET, metabolic equivalent task; EQ-5D, Euro Quality of Life-5 Dimension Standard Frail: if ≥3 criteria are fulfilled, Pre-frail: 1 or 2 criteria are fulfilled, Robust: none of the criteria are fulfilled.

**Table 2 nutrients-17-02977-t002:** Baseline characteristics of participants in the Non-DS, Other DS, and Vitamin C supplements group.

Characteristics	Non-DS (*n* = 1517, *N* = 4.01)	Other DS (*n* = 1227, *N* = 3.27)	Vitamin C (*n* = 75, *N* = 0.21)	*p* Value *	*p* Value †
Age, years	73.43 (0.18)	72.61 (0.18)	72.83 (0.60)	0.3268	0.7207
65–74 years, %	54.8 (1.8)	62.4 (1.9)	64.8 (6.7)	0.1587	0.7279
≥75 years, %	45.2 (1.8)	37.6 (1.9)	35.2 (6.7)		
Sex, women, %	51.4 (1.5)	60.6 (1.5)	40.5 (6.1)	0.0885	0.0017
Education level ≤ 9 years, %	74.4 (1.6)	64.3 (2.0)	53.8 (7.2)	0.0015	0.1256
The lowest 25% quartile of income, %	50.0 (1.9)	41.3 (2.0)	38.6 (6.7)	0.1025	0.6916
Living alone, %	19.7 (1.1)	19.2 (1.4)	23.3 (5.4)	0.4859	0.4474
Current smoker, %	10.7 (1.0)	7.3 (0.9)	8.9 (3.5)	0.6248	0.6526
Frequent alcohol drinking, %	18.5 (1.1)	14.8 (1.1)	16.9 (6.0)	0.8045	0.7226
Total MET, kcal/week	399.13 (25.90)	530.14 (31.46)	672.32 (153.25)	0.0717	0.3526
Body mass index, kg/m^2^	24.13 (0.09)	24.00 (0.10)	24.06 (1.35)	0.8689	0.8590
Underweight (BMI < 18.5)	2.7 (0.5)	2.2 (0.4)	0.5 (0.5)	0.5548	0.6742
Normal (18.5 ≤ BMI < 23)	33.1 (1.4)	36.5 (1.6)	34.2 (6.6)		
Overweight (23 ≤ BMI < 25)	26.1 (1.3)	26.8 (1.5)	30.2 (6.3)		
Obesity (25 ≥ BMI)	38.1 (1.4)	34.4 (1.6)	35.1 (6.0)		
Number of comorbidities ≥ 3, %	20.4 (1.3)	16.3 (1.2)	16.7 (5.0)	0.5127	0.9402
Nutritional intake					
Total energy, kcal	1567.05 (21.10)	1627.05 (21.61)	1678.22 (87.69)	0.2234	0.5661
Carbohydrate, g	269.54 (3.37)	270.96 (3.53)	287.12 (15.97)	0.2815	0.3167
Protein, g	51.89 (0.87)	57.23 (0.98)	57.33 (3.27)	0.1110	0.9786
Fat, g	25.18 (0.68)	30.00 (0.76)	25.95 (1.89)	0.6975	0.0470
Vitamin C, mg					
Food	52.70 (2.24)	57.25 (1.87)	56.38 (5.63)	0.5297	0.8797
Food + Supplements	52.70 (2.24)	102.52 (4.10)	1447.34 (177.95)	<0.0001	<0.0001
Frailty					
Robust, %	16.6 (1.3)	20.8 (1.4)	30.4 (5.6)	0.0109	0.1915
Pre-frail, %	66.1 (1.4)	66.6 (1.6)	60.6 (6.3)		
Frail, %	17.3 (1.2)	12.6 (1.1)	9.0 (3.8)		

Abbreviation: DS, dietary supplements; *n*, unweighted sample size; *N*, weighted sample size in millions; MET, metabolic equivalent task; BMI, body mass index. Comorbidities included hypertension, cardiovascular disease, stroke, diabetes mellitus and cancers. Cancers included stomach, colon, liver, breast, cervical, lung, and other cancers. * Comparisons of Non-DS and Vitamin C. † Comparisons of Other DS and Vitamin C.

**Table 3 nutrients-17-02977-t003:** Association between intake of vitamin C supplements and frailty.

	Crude OR (95% CI)	Adjusted OR (95% CI)	Adjusted Probability % (95% CI)	Absolute Risk Difference % (95% CI)	*p*-Value
Non-DS vs. Vitamin C	0.46 (0.18–1.17)	0.65 (0.26–1.66)	16.2 (14.1–18.3) vs. 11.6 (2.8–20.3)	−4.6 (−13.6 to +4.3)	0.308
Other DS vs. Vitamin C	0.68 (0.26–1.75)	0.81 (0.26–2.53)	13.6 (11.5–15.8) vs. 11.6 (2.8–20.3)	−2.1 (−11.2 to +7.1)	0.656

Non-DS are participants who did not consume any type of dietary supplements (DS). Values are presented as odds ratios (ORs) with 95% confidence intervals (CIs), adjusted predicted probabilities, and absolute risk differences derived from survey-weighted logistic regression models. Multivariable models were adjusted for age, sex, education level, income level, living alone, body mass index, alcohol consumption, smoking status, number of chronic diseases, total energy intake, and protein intake.

**Table 4 nutrients-17-02977-t004:** Frailty according to of daily vitamin C intake (food and supplements combined).

Daily Intake of Vitamin C	<100 mg (*n* = 2119, *N* = 5.61)	100–999 mg (*n* = 628, *N* = 1.68)	1000–1999 mg (*n* = 53, *N* = 0.15)	≥2000 mg (*n* = 16, *N* = 0.04)	*p* for Trend
Mean vitamin C intake, mg					
Food	27.31 (26.08–28.60)	84.00 (77.07–91.55)	41.84 (31.16–56.17)	25.88 (13.58–49.32)	<0.001
Food+ Supplements	28.70 (27.39–30.07)	175.73 (1.02) (168.14–183.67)	1066.89 (1049.45–1084.63)	2974.05 (2252.13–3927.39)	<0.001
Frailty status, %					
Robust	16.1 (1.1)	26.7 (2.1)	26.7 (6.4)	30.7 (10.7)	<0.001
Pre-frail	67.1 (1.2)	63.4 (2.3)	63.0 (7.2)	62.9 (11.3)	
Frail	16.8 (1.1)	9.9 (1.2)	10.3 (4.1)	6.4 (6.2)	
OR	1 (ref)	0.79 (0.57–1.08)	0.79 (0.31–2.02)	0.46 (0.05–3.68)	0.120

Abbreviation: *n*, unweighted sample size; *N*, weighted sample size in millions. Values are presented as weighted means (95% confidence interval) or weighted percentages (standard error). Daily intake of vitamin C is estimated from both dietary and supplement sources. Odds ratios (ORs) and 95% confidence intervals (CIs) for frailty are obtained from survey-weighted logistic regression models, with the lowest intake group (<100 mg/day) as the reference. Multivariable models are adjusted for age, sex, education level, household income, living alone, body mass index, alcohol consumption, smoking status, number of chronic diseases, and total energy and protein intake.

## Data Availability

The original dataset used in this study can be found in the Korea National Health and Nutrition Examination Survey repository (https://knhanes.kdca.go.kr/) (accessed on 28 December 2024).

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
