# Peer review of "No Significant Association Between Vitamin C Supplements and Frailty in Korean Older Adults: A Cross-Sectional Analysis of the 2018–2019 Korea National Health and Nutrition Examination Survey"

_nutrients, 2025, doi:10.3390/nu17182977_

Round 1

Reviewer 1 Report

Comments and Suggestions for Authors

The authors analyzed cross-sectional survey data to evaluate associations between vitamin
C supplementation and frailty in Koreans aged ≥65 years. They report significant associations between Vitamin C supplementation and lower prevalence of frailty; with a dose-response relationship based on frequency and dose of supplementation. The topic is of public health importance. Suggestions:

1) Abstract conclusion (Lines 28-29): "Vitamin C supplementation over a sufficient period 28
based on a regular diet is expected to help manage frailty.
" This is an overstatement for these cross-sectional data- please delete or rephrase to avoid making any causal inference.

2) Please use and cite the STROBE reporting guideline and attach a completed STROBE checklist in the appendix.

3) Please quantify missing covariate data and describe how it was handled in analyses; e.g. was a sensitivity analysis using MICE/multiple imputation done/considered?

4) Consider presenting forest plots for tables 3 to 5

5) Consider evaluating associations with individual frailty components as the dependent variable

6) In the limitations para, please discuss the possibility of reverse causation.

7) Another major limitation of these cross-sectional analyses for nutritional research is not accounting for other dietary factors that could potentially confound associations... including dietary Vit C and other dietary factors/medications/supplements... please consider adjusting for more variables if available and in the limitations para discuss this key limitation

Author Response

Reviewer 1

We sincerely thank the reviewers for their valuable and constructive comments, which have greatly improved our manuscript.

During the revision process, we carefully considered the suggestions from all three reviewers. In particular, Reviewer 3 raised concerns regarding the sample selection and comparator groups. In response, we restructured the exposure groups and re-ran the analyses accordingly. As a result, the revised manuscript differs substantially from the originally submitted version, with notable changes in the Results, Discussion, and Conclusions sections.

We kindly ask for the reviewers’ understanding regarding these broader revisions. Although these changes extend beyond what Reviewers 1 and 2 may have anticipated, we believe they substantially strengthen the validity and clarity of our findings.

Below, we provide a detailed, point-by-point response to each reviewer’s comments. All changes have been highlighted in the revised manuscript, and modifications are marked in red color for clarity.

Comments and Suggestions for Authors

The authors analyzed cross-sectional survey data to evaluate associations between vitamin C supplementation and frailty in Koreans aged ≥65 years. They report significant associations between Vitamin C supplementation and lower prevalence of frailty; with a dose-response relationship based on frequency and dose of supplementation. The topic is of public health importance. Suggestions:

Comments 1: Abstract conclusion (Lines 28-29): "Vitamin C supplementation over a sufficient period 28 based on a regular diet is expected to help manage frailty." This is an overstatement for these cross-sectional data- please delete or rephrase to avoid making any causal inference.

Response 1: We fully agree with the reviewer’s concern. In addition to revising the expression to avoid any causal inference, we note that the overall findings of the manuscript have changed after re-analysis with restructured groups, as suggested during the revision. Accordingly, the Abstract conclusion has been modified to reflect the updated results, which no longer indicate a significant association. The revised sentence now reads “Vitamin C supplementation was not significantly associated with frailty in Korean older adults.” (Lines 28)

Comments 2: Please use and cite the STROBE reporting guideline and attach a completed STROBE checklist in the appendix.

Response 2: We thank the reviewer for this important suggestion. In the revised manuscript, we have now cited the STROBE reporting guideline in the Methods section (Lines 95–97). In addition, a completed STROBE checklist has been prepared and is provided as Supplementary Appendix A

Comments 3: Please quantify missing covariate data and describe how it was handled in analyses; e.g. was a sensitivity analysis using MICE/multiple imputation done/considered?

Response 3: We thank the reviewer for this helpful suggestion. In the revised manuscript, we have quantified the extent of missing data. Among the 3,395 participants aged ≥65 years in KNHANES 2018–2019, 315 were excluded due to missing values for dietary supplement intake, and 140 were excluded due to missing values in frailty components, sociodemographic, or nutrition-related variables. In addition, 18 individuals with extreme values for physical activity were excluded. As a result, 2,922 participants were included in the final analysis. These details are now described in the Methods – Study Population section (Lines 83–88) and depicted in Figure1. As shown in the flow chart (Figure 1), participants with missing values in key variables (dietary supplement intake, frailty components, or major covariates) were excluded from the analyses. In the multivariable logistic regression analyses, covariates with a small proportion of missing values were included without imputation, because the proportion of missing data was minimal. Therefore, participants with missing values for a given covariate were excluded only from the specific model that included that covariate (complete-case approach). Given the low frequency of missing data, this approach was considered unlikely to introduce meaningful bias.

Comments 4: Consider presenting forest plots for tables 3 to 5

Response 4: We appreciate the reviewer’s helpful suggestion. We generated forest plots for the regression analyses corresponding to Tables 3–4. However, these figures did not reveal any clear dose–response pattern beyond what was already presented in the tables. Therefore, we decided not to add them to the main text, but we have provided them as Supplementary Figures for completeness.

Comments 5:  Consider evaluating associations with individual frailty components as the dependent variable

Response 5: We thank the reviewer for this constructive suggestion. In the revised manuscript, we additionally examined the associations between vitamin C supplementation and each of the individual frailty components (weight loss, exhaustion, weakness, slowness, and low physical activity). The results are presented in Supplementary Table S1. No consistent associations were observed for specific components, which is in line with the overall null findings in the main analyses. This addition is described in the Methods – Frailty Definition and briefly mentioned in the Results section (Lines 119-125).

Comments 6: In the limitations para, please discuss the possibility of reverse causation.

Response 6: We appreciate the reviewer’s suggestion. In the revised manuscript, we have emphasized the potential for reverse causation in two places. First, in the Limitations section, we clearly stated that the cross-sectional design precludes causal inference (Lines 308-309). Second, in the Discussion (Lines 279-281), we explicitly noted that some prior studies showing protective associations may themselves have been affected by reverse causation, as frailer individuals might have been more motivated to take supplements. We believe this dual mention sufficiently addresses the reviewer’s concern.

Comments 7: Another major limitation of these cross-sectional analyses for nutritional research is not accounting for other dietary factors that could potentially confound associations... including dietary Vit C and other dietary factors/medications/supplements... please consider adjusting for more variables if available and in the limitations para discuss this key limitation

Response 7: We appreciate this important comment. In the revised analysis, we additionally adjusted for protein intake, which was available in KNHANES and serves as a key indicator of diet quality. Unfortunately, detailed data on other dietary components, micronutrients, or the use of additional dietary supplements and medications were not sufficiently available to be incorporated into the models. We have expanded the Discussion – Limitations section to acknowledge the potential for residual confounding. Specifically, we now note that other dietary components, micronutrient intake, medications, and the use of other supplements were not fully captured and may represent a potential source of residual confounding in this analysis (Lines 320–327).

Reviewer 2 Report

Comments and Suggestions for Authors

This manuscript investigates the association between vitamin C supplementation and frailty among older Korean adults using data from the KNHANES 2018–2019 survey. The study addresses an important and timely public health issue, as frailty is a major determinant of quality of life and healthcare burden in aging societies. The manuscript is generally well-structured and the analysis is appropriate. However, several minor changes should be done.

Abstract

The phrasing “marginally associated” for a non-significant result (OR 0.59, CI 0.34–1.01) is misleading and should be revised to reflect lack of statistical significance.

The conclusion “expected to help manage frailty” is somewhat speculative; please emphasize association rather than causality.

Materials and Methods

The modified definition of frailty should be more clearly justified, since deviations from the standard Fried criteria may limit comparability with other studies.

The rationale for dichotomizing vitamin C intake duration into <1 year and ≥1 year is not clearly explained; please clarify.

Self-reported supplement use is a major limitation and should be acknowledged here, not only in the Discussion.

Many subgroup analyses were conducted, but no adjustment for multiple comparisons is mentioned. (eg Bonferroni, Tukey, ) Please discuss this issue.

Results

Some results are overemphasized despite wide confidence intervals (e.g., ≥2000 mg group, OR 0.36, CI 0.05–2.69). Please interpret cautiously.

Non-significant results are sometimes described as “marginal.” Please use more precise language.

Discussion

The tone of the discussion is at times overstated, particularly when suggesting causal implications from cross-sectional data. Please reframe findings strictly as associations.

Reverse causality (frail individuals may avoid supplements) should be explicitly discussed.

Limitations such as recall bias, residual confounding, and self-reported supplement intake should be emphasized more strongly.

Author Response

Reviewer 2

We sincerely thank the reviewers for their valuable and constructive comments, which have greatly improved our manuscript.

During the revision process, we carefully considered the suggestions from all three reviewers. In particular, Reviewer 3 raised concerns regarding the sample selection and comparator groups. In response, we restructured the exposure groups and re-ran the analyses accordingly. As a result, the revised manuscript differs substantially from the originally submitted version, with notable changes in the Results, Discussion, and Conclusions sections.

We kindly ask for the reviewers’ understanding regarding these broader revisions. Although these changes extend beyond what Reviewers 1 and 2 may have anticipated, we believe they substantially strengthen the validity and clarity of our findings.

Below, we provide a detailed, point-by-point response to each reviewer’s comments. All changes have been highlighted in the revised manuscript, and modifications are marked in red color for clarity.

Comments and Suggestions for Authors

This manuscript investigates the association between vitamin C supplementation and frailty among older Korean adults using data from the KNHANES 2018–2019 survey. The study addresses an important and timely public health issue, as frailty is a major determinant of quality of life and healthcare burden in aging societies. The manuscript is generally well-structured and the analysis is appropriate. However, several minor changes should be done.

Abstract>

Comments 1:  The phrasing “marginally associated” for a non-significant result (OR 0.59, CI 0.34–1.01) is misleading and should be revised to reflect lack of statistical significance.

Response 1: We fully agree with the reviewer’s point. In the revised manuscript, we have removed the term “marginally associated” and replaced it with wording that more accurately reflects the lack of statistical significance. The revised sentence now reads:

“Vitamin C supplementation was not significantly associated with frailty.” (Lines 28)

Comments 2:  The conclusion “expected to help manage frailty” is somewhat speculative; please emphasize association rather than causality.

Response 2: We thank the reviewer for this valuable comment. In the revised manuscript, we have rephrased the conclusion to avoid causal language and to emphasize only the observed association. The Abstract and Conclusion sections now state:

“Vitamin C supplementation was not significantly associated with frailty. “(Lines 28)

Materials and Methods>

Comments 3: The modified definition of frailty should be more clearly justified, since deviations from the standard Fried criteria may limit comparability with other studies.

Response 3: We appreciate this important comment. In the revised manuscript, we have expanded the Methods – Frailty Definition section to provide a clearer justification for the modified criteria. Specifically, we explained that the standard Fried phenotype could not be applied directly due to the variables available in KNHANES. Therefore, we used validated proxy measures that have been adopted in previous Korean studies, including perceived stress as a proxy for exhaustion and the EQ-5D mobility item as a proxy for slowness. Handgrip strength was assessed using AWGS 2019 cut-offs, which are widely accepted for Asian populations. References to prior KNHANES-based frailty studies using these operational definitions have been added (Lines 107–125).

We also acknowledged in the Discussion – Limitations section that this modified definition may limit comparability with studies using the original Fried criteria. (Lines 313–315).

Comments 4: The rationale for dichotomizing vitamin C intake duration into <1 year and ≥1 year is not clearly explained; please clarify.

Response 4: We appreciate the reviewer’s comment. In the original submission, we presented an analysis stratified by vitamin C intake duration (<1year vs ≥1year). However, following re-analysis with restructured exposure groups to address Reviewer 3’s concerns regarding sample selection, this duration-based comparison was not included in the revised analyses.

Comments 5: Self-reported supplement use is a major limitation and should be acknowledged here, not only in the Discussion

Response 5: We appreciate the reviewer’s comment. In the revised manuscript, we have explicitly acknowledged in the Methods – Exposure Assessment section that vitamin C supplement use was assessed through self-report rather than objective verification. The revised text now states:

“Because supplement use was assessed by self-report rather than objective verification (e.g., pill counts, prescription records, or biomarkers), potential recall bias and misclassification cannot be ruled out.” (Lines 140–142).

This limitation remains highlighted in the Discussion section as well (Lines 309–311).

Comments 6: Many subgroup analyses were conducted, but no adjustment for multiple comparisons is mentioned. (eg Bonferroni, Tukey, ) Please discuss this issue.

Response 6: We appreciate the reviewer’s observation. In the revised analysis, subgroup analyses were not retained, as the exposure groups were restructured to address Reviewer 3’s concerns and the focus was placed on the main associations. Therefore, the issue of multiple comparisons is no longer applicable in the updated manuscript.

Results>

Comments 7: Some results are overemphasized despite wide confidence intervals (e.g., ≥2000 mg group, OR 0.36, CI 0.05–2.69). Please interpret cautiously.

Response 7: We thank the reviewer for this important point. We agree that the results for higher intake groups, particularly ≥2000 mg/day, were based on very small numbers and accompanied by wide confidence intervals. These findings have limited interpretability and do not provide meaningful evidence of a dose–response association. Therefore, we have avoided overemphasizing these results and did not add further discussion in the main text.

Comments 8: Non-significant results are sometimes described as “marginal.” Please use more precise language.

Response 8: We appreciate the reviewer’s observation. We have carefully reviewed the manuscript and removed all instances of the term “marginal” to describe non-significant results. These results are now consistently reported as “not statistically significant” to ensure precise language.

Discussion>

Comments 9: The tone of the discussion is at times overstated, particularly when suggesting causal implications from cross-sectional data. Please reframe findings strictly as associations.

Response 9: We fully agree with the reviewer. In the revised manuscript, we carefully revised the Discussion and Conclusion sections to remove any causal wording and to emphasize that our results reflect associations only. For example, phrases such as “expected to help manage frailty” were replaced with more cautious wording (e.g., “was not significantly associated with frailty”). These revisions ensure that the tone of the manuscript is appropriately cautious and consistent with the cross-sectional design.

Comments 10:  Reverse causality (frail individuals may avoid supplements) should be explicitly discussed.

Response 10: We appreciate the reviewer’s suggestion. In the revised manuscript, we have emphasized the potential for reverse causation in two places. First, in the Limitations section, we clearly stated that the cross-sectional design precludes causal inference (Lines 308-310). Second, in the Discussion (Lines 279-281)., we explicitly noted that some prior studies showing protective associations may themselves have been affected by reverse causation, as frailer individuals might have been more motivated to take supplements. We believe this dual mention sufficiently addresses the reviewer’s concern.

Comments 11: Limitations such as recall bias, residual confounding, and self-reported supplement intake should be emphasized more strongly.

Response 11: We appreciate the reviewer’s suggestion. In the revised manuscript, we strengthened the Discussion – Limitations section to emphasize these issues more clearly. We explicitly noted that supplement use was based on self-report and may be subject to recall bias and misclassification, and we highlighted the possibility of residual confounding despite adjustment for major covariates. These revisions ensure that the limitations are more prominently acknowledged (Lines 309-311, 320-327).

Reviewer 3 Report

Comments and Suggestions for Authors

Dear editors and authors:      

        It is a great honor and pleasure for me to be invited as the reviewer for this important work entitled “Association between Vitamin C Supplements and Frailty in Korean Older Adults: Analysis of the 2018–2019 Korea National Health and Nutrition Examination Survey”. Hyoeun Kim and co-authors comprehensively reviewed the potential effect of Vitamin C Supplements on Frailty in Korean Older Adults. This cross-sectional analysis uses KNHANES 2018–2019 to examine whether vitamin C supplement use among adults ≥65 years is associated with lower frailty prevalence. The final analytic sample contrasts a “vitamin C supplement” group (n=179) with a “non–dietary-supplement (non-DS)” group (n=1,530) selected from 2,949 eligible older adults. The authors use a modified Fried phenotype constructed from KNHANES items (weight loss, perceived stress as exhaustion, handgrip weakness, EQ-5D mobility for slowness, and low physical activity), classifying frailty as meeting ≥3 criteria. They report lower odds of frailty among vitamin C users (OR 0.40, 95% CI 0.18–0.84), with a dose category of 1,000–1,999 mg/day (combined diet+supplement) showing OR 0.37 (0.16–0.84) and intake ≥1 year showing OR 0.26 (0.10–0.65). This study topic is novel and interesting, attributing to their team’s long-term efforts and contributions in the scientific field. I have a number of comments concerning this study:

Major comments

  1. Sample selection and comparator choice introduce selection bias.

From 2,949 older adults, the analysis contrasts vitamin C supplement users (n=179) only against respondents who reported no dietary supplement use (n=1,530), effectively excluding users of other supplements who did not take vitamin C. This design can inflate differences due to “any supplement user vs none” behaviors and socioeconomic factors, rather than vitamin C per se. Please justify this exclusion strategy and, preferably, re-analyze including all participants with exposure coded as (a) total vitamin C intake (food+supplement, continuous), and (b) supplement categories (vitamin C only, vitamin C + other supplements, other supplements without vitamin C, none). Provide a directed acyclic graph to clarify potential collider bias introduced by conditioning on supplement use.

  1. Exposure measurement needs clarification and stronger modeling.

The abstract and methods state categories (<100, 100–999, 1,000–1,999, ≥2,000 mg/day) derived from diet+supplements, but the primary contrast is “vitamin C supplement users” vs “non-DS”. Clarify whether vitamin C users could also take other supplements and whether dose categories include the non-DS group (many of whom still consume vitamin C from food). Please add:

  • Numbers (n, weighted %) in each dose category and by duration strata.
  • A model treating vitamin C intake as a continuous variable (e.g., log-mg) using restricted cubic splines to test nonlinearity and avoid arbitrary cut-points.
  • Sensitivity analyses using intake from food only vs supplement only vs combined, as claims about dose largely reflect supplemental intake (mean intake 1,354.7 mg in vitamin C users vs 52.5 mg in non-DS).
  1. Frailty phenotype construction and potential misclassification.

The modified Fried phenotype maps “slowness” to EQ-5D mobility and “exhaustion” to perceived stress, and defines weakness via dominant-hand grip thresholds (<28 kg men, <18 kg women). Please provide validation references for each proxy in KNHANES and report sensitivity analyses:

  • Use alternative handgrip definitions (e.g., non-dominant or best of both hands) and AWGS thresholds to check robustness.
  • Test a version excluding the physical activity component to reduce potential overadjustment/construct overlap if physical activity is later used as a covariate.
  • Present prevalence of each component by exposure to evaluate differential misclassification.
  1. Confounding and healthy-user behavior.

Baseline differences show that vitamin C users were younger and had higher energy/protein/fat intake and higher education/income—classic “healthy user” patterns. Although models adjust for some factors (age, sex, education, income, living arrangement, BMI, smoking, drinking, chronic disease, energy intake), this may be insufficient. Please:

  • Adjust for diet quality (e.g., protein intake, fruit/vegetable servings or a diet quality index) and physical activity (if not part of the frailty construct in that model), or justify exclusion to avoid overadjustment.
  • Consider propensity score weighting/matching including socioeconomic and lifestyle variables to reduce imbalance.
  • Provide E-values for the main ORs to quantify robustness to unmeasured confounding.
  1. Complex survey analysis details.

You state accounting for kstrata, PSU, and weights and using STATA 15 SE. Confirm explicitly that logistic models were estimated with the survey design (e.g., svy: commands) and report the weighted prevalence and design effects. Provide the weight variable name and how 24-hour recall weights were harmonized with examination/interview weights, if applicable.

  1. Reverse causation and temporality.

The cross-sectional design prevents establishing that supplement use preceded frailty. The duration analysis (≥1 year vs <1 year) is suggestive but vulnerable to survivorship and healthy-user biases. Tone down causal language throughout and consider a negative-control exposure (e.g., a supplement unlikely to influence frailty) as a falsification test if available, at least discuss healthy-user bias more explicitly.

  1. Precision and multiplicity.

The highest dose strata (≥2,000 mg/day) likely contain very small numbers with wide CIs; present exact n, weighted %, and instability diagnostics. When presenting multiple dose and duration comparisons and subgroup analyses (age, sex, income), report interaction p-values and address multiplicity (e.g., false discovery rate or emphasize estimation over hypothesis testing).

  1. Dietary intake assessment and measurement error.

A single 24-hour recall is noisy for individual vitamin C intake. You note attempts to match supplement reporting with recall, but serum vitamin C was unavailable. Please expand on expected direction of bias (likely toward the null for diet alone) and consider regression calibration or sensitivity analyses to assess misclassification.  

  1. Outcome communication—absolute risks.

Given the relatively low frailty prevalence (5.3% in vitamin C users, 17.2% in non-DS), provide absolute risk differences or adjusted marginal probabilities alongside ORs to aid clinical interpretation.

Minor comments

  • Abstract/Conclusions: Rephrase to avoid causal wording (e.g., “is associated with lower prevalence” rather than “helps manage”). The abstract already uses associative language in places; ensure consistency throughout.
  • Methods clarity: Add a participant-flow figure with explicit counts at each exclusion, including numbers of “other supplement users” removed from the comparator.
  • Reporting: State the Korean RNI (100 mg/day) and Korean UL (2,000 mg/day) in the Methods where dose categories are introduced, with references; the rationale is now scattered across sections. Ensure the 1,000 mg “UL by WHO” phrasing is accurate or corrected.
  • Tables: In Table 2 and dose/duration tables, add survey-weighted means/percentages with 95% CIs. Provide footnotes defining all abbreviations and the exact survey weight used.
  • Language/formatting: A handful of phrasing issues (e.g., “adversely linear” → “inverse linear”) and spacing/typography should be polished during revision.
  • Ethics/data availability: IRB approvals and data availability statements are appropriate; keep accession links stable.

The dataset is nationally representative, and the authors appropriately mention complex survey design. However, key aspects of exposure definition, sample selection, potential healthy-user/selection biases, and interpretation need substantial revision before the paper is reliable for publication. I recommend major revision.

Author Response

Reviewer 3

We sincerely thank the reviewers for their valuable and constructive comments, which have greatly improved our manuscript.

During the revision process, we carefully considered the suggestions from all three reviewers. In particular, Reviewer 3 raised concerns regarding the sample selection and comparator groups. In response, we restructured the exposure groups and re-ran the analyses accordingly. As a result, the revised manuscript differs substantially from the originally submitted version, with notable changes in the Results, Discussion, and Conclusions sections.

We kindly ask for the reviewers’ understanding regarding these broader revisions. Although these changes extend beyond what Reviewers 1 and 2 may have anticipated, we believe they substantially strengthen the validity and clarity of our findings.

Below, we provide a detailed, point-by-point response to each reviewer’s comments. All changes have been highlighted in the revised manuscript, and modifications are marked in red color for clarity.

Comments and Suggestions for Authors

Dear editors and authors:     

 It is a great honor and pleasure for me to be invited as the reviewer for this important work entitled “Association between Vitamin C Supplements and Frailty in Korean Older Adults: Analysis of the 2018–2019 Korea National Health and Nutrition Examination Survey”. Hyoeun Kim and co-authors comprehensively reviewed the potential effect of Vitamin C Supplements on Frailty in Korean Older Adults. This cross-sectional analysis uses KNHANES 2018–2019 to examine whether vitamin C supplement use among adults ≥65 years is associated with lower frailty prevalence. The final analytic sample contrasts a “vitamin C supplement” group (n=179) with a “non–dietary-supplement (non-DS)” group (n=1,530) selected from 2,949 eligible older adults. The authors use a modified Fried phenotype constructed from KNHANES items (weight loss, perceived stress as exhaustion, handgrip weakness, EQ-5D mobility for slowness, and low physical activity), classifying frailty as meeting ≥3 criteria. They report lower odds of frailty among vitamin C users (OR 0.40, 95% CI 0.18–0.84), with a dose category of 1,000–1,999 mg/day (combined diet+supplement) showing OR 0.37 (0.16–0.84) and intake ≥1 year showing OR 0.26 (0.10–0.65). This study topic is novel and interesting, attributing to their team’s long-term efforts and contributions in the scientific field. I have a number of comments concerning this study:

Major comments

1.Sample selection and comparator choice introduce selection bias.

Comments 1: From 2,949 older adults, the analysis contrasts vitamin C supplement users (n=179) only against respondents who reported no dietary supplement use (n=1,530), effectively excluding users of other supplements who did not take vitamin C. This design can inflate differences due to “any supplement user vs none” behaviors and socioeconomic factors, rather than vitamin C per se. Please justify this exclusion strategy and, preferably, re-analyze including all participants with exposure coded as (a) total vitamin C intake (food+supplement, continuous), and (b) supplement categories (vitamin C only, vitamin C + other supplements, other supplements without vitamin C, none). Provide a directed acyclic graph to clarify potential collider bias introduced by conditioning on supplement use.

Response 1: We thank the reviewer for highlighting the issue of potential collider bias when conditioning on supplement use. In this context, socioeconomic status and underlying health status may both influence supplement use, and conditioning on supplement use could therefore create a spurious association between these factors and frailty (i.e., collider bias). To minimize this problem, rather than treating all supplement users as a single category, we restructured the exposure groups to focus on two comparisons: (i) vitamin C only vs no supplement use, and (ii) vitamin C only vs other supplements without vitamin C. This approach was intended to reduce the risk of collider bias by avoiding undue conditioning on overall supplement use. We clarified this rationale in the revised Methods – Exposure Assessment section (Lines 133-139).

  1. Exposure measurement needs clarification and stronger modeling.

Comments 2: The abstract and methods state categories (<100, 100–999, 1,000–1,999, ≥2,000 mg/day) derived from diet+supplements, but the primary contrast is “vitamin C supplement users” vs “non-DS”. Clarify whether vitamin C users could also take other supplements and whether dose categories include the non-DS group (many of whom still consume vitamin C from food). Please add:

Response 1: The dose categories (<100, 100–999, 1,000–1,999, and ≥2,000 mg/day) were defined based on total vitamin C intake from both food and supplements. Thus, participants in the non-supplement group (non-DS) were also included in the <100 mg/day category through their dietary intake alone.

Comments 2-1: Numbers (n, weighted %) in each dose category and by duration strata.

Response 2-1:  As suggested, we have added both unweighted numbers and survey-weighted number for each vitamin C intake category and duration stratum. These are now presented in Tables 2, and 4 of the revised manuscript.

Comments 2-2: A model treating vitamin C intake as a continuous variable (e.g., log-mg) using restricted cubic splines to test nonlinearity and avoid arbitrary cut-points.

Response 2-2: We appreciate the reviewer’s suggestion. Daily vitamin C intake from food and supplements showed a markedly skewed distribution and was therefore log-transformed to approximate normality. Quartile cut-off points derived from the transformed values were then used to classify participants into four intake categories (<100, 100–999, 1,000–1,999, and ≥2,000 mg/day), which were subsequently applied in the regression analyses. We did not implement spline-based models. The number of participants in the higher intake categories was very small (e.g., only 16 participants in the ≥2,000 mg/day group), which would have led to unstable estimates and wide confidence intervals in spline analyses. For this reason, we considered the categorical approach more reliable for the current dataset.

Comments 2-3: Sensitivity analyses using intake from food only vs supplement only vs combined, as claims about dose largely reflect supplemental intake (mean intake 1,354.7 mg in vitamin C users vs 52.5 mg in non-DS).

Response 2-3: We appreciate the reviewer’s suggestion. To clarify the contribution of dietary versus supplemental sources, we revised Table 4 to separately present vitamin C intake from food alone and from food plus supplements. We have noted this distinction explicitly in the revised Results. (Lines 234-237)

  1. Frailty phenotype construction and potential misclassification.

Comments 3: The modified Fried phenotype maps “slowness” to EQ-5D mobility and “exhaustion” to perceived stress, and defines weakness via dominant-hand grip thresholds (<28 kg men, <18 kg women). Please provide validation references for each proxy in KNHANES and report sensitivity analyses:

Response 3: We thank the reviewer for this helpful suggestion. In the revised manuscript, we analyzed the prevalence of each frailty component according to vitamin C supplementation status. The results are presented in Supplementary Table S1. In these supplementary analyses, each frailty component was modeled as a separate dependent variable, and the results were generally consistent. We acknowledge that these operational definitions may introduce misclassification; however, no consistent pattern across components was observed, suggesting that differential misclassification by exposure is unlikely. In addition, prior Korean studies have demonstrated the construct validity of these proxies and reported consistent findings. (Lines 119-125)

Comments 3-1: Use alternative handgrip definitions (e.g., non-dominant or best of both hands) and AWGS thresholds to check robustness.

Response 3-1: We thank the reviewer for this suggestion. In KNHANES, handgrip strength was measured with up to three trials on the dominant hand (six trials if participants were ambidextrous). Thus, for the main analysis we defined weakness using the maximum grip strength of the dominant hand, applying the AWGS 2019 cut-offs (<28 kg for men, <18 kg for women). For ambidextrous participants, both hands were assessed and the stronger value was used, which is consistent with a “best of both hands” approach in this subgroup (table1). Unfortunately, alternative definitions (e.g., best of both hands for all participants) were not available in the dataset.

Comments 3-2: Test a version excluding the physical activity component to reduce potential overadjustment/construct overlap if physical activity is later used as a covariate.

Response 3-2: We appreciate the reviewer’s concern. In our analyses, physical activity was not included as a covariate in the multivariable models. Therefore, there was no risk of overadjustment or construct overlap, and we retained the standard five-component definition of frailty for consistency with prior KNHANES-based studies. This has been clarified in the revised Methods (Lines 185-187)

Comments 3-3: Present prevalence of each component by exposure to evaluate differential misclassification.

Response 3-3: We thank the reviewer for this helpful suggestion. As requested, we analyzed the prevalence of each frailty component according to vitamin C supplementation status. The results are presented in Supplementary Table S1. No consistent differences across components were observed, suggesting that differential misclassification by exposure is unlikely.

  1. Confounding and healthy-user behavior.

Comments 4: Baseline differences show that vitamin C users were younger and had higher energy/protein/fat intake and higher education/income—classic “healthy user” patterns. Although models adjust for some factors (age, sex, education, income, living arrangement, BMI, smoking, drinking, chronic disease, energy intake), this may be insufficient. Please:

Response 4: As noted earlier, to minimize selection bias we excluded the vitamin C + other supplements group, which had shown strong imbalance. After this reclassification, the baseline differences between groups were markedly reduced, as shown in Table 2. However, because the number of vitamin C–only users was relatively small, the standard errors increased despite improved comparability. We adjusted for additional covariates including protein intake as a marker of diet quality, but we acknowledge that residual confounding related to “healthy-user” behavior cannot be fully excluded. This has been emphasized in the Discussion – Limitations section. (Lines 320-327)

Comments 4-1: Adjust for diet quality (e.g., protein intake, fruit/vegetable servings or a diet quality index) and physical activity (if not part of the frailty construct in that model), or justify exclusion to avoid overadjustment.

Response 4-1: As suggested, we additionally included protein intake as a covariate in the multivariable models to account for diet quality. Physical activity was not entered as a covariate because it is already part of the frailty construct, and inclusion could lead to overadjustment. This has been clarified in the revised Methods – Statistical Analysis section (Lines 185-187, 320–327).

Comments 4-2: Consider propensity score weighting/matching including socioeconomic and lifestyle variables to reduce imbalance.

Response 4-2: We appreciate the reviewer’s valuable suggestion. We explored the feasibility of applying propensity score weighting and matching. However, because the exposure was classified into three groups (vitamin C only, other supplements without vitamin C, and non-users) rather than a simple binary contrast, implementing stable propensity score methods was not straightforward. Instead, we used multivariable logistic regression with adjustment for major sociodemographic, lifestyle, and nutritional covariates, which we considered more appropriate for the current dataset.

Comments 4-3: Provide E-values for the main ORs to quantify robustness to unmeasured confounding.

Response 4-3: Given that most associations were not statistically significant, the E-value results provide limited additional insight. For example, the E-values were 1.6 for the comparisons of vitamin C–only users versus non-users and versus other supplement users, and 3.7 for the ≥2000 mg/day intake group. These findings indicate that relatively modest unmeasured confounding could account for the observed associations, but they should not be interpreted as evidence of strong or hidden effects.

  1. Complex survey analysis details.

Comments 5: You state accounting for kstrata, PSU, and weights and using STATA 15 SE. Confirm explicitly that logistic models were estimated with the survey design (e.g., svy: commands) and report the weighted prevalence and design effects. Provide the weight variable name and how 24-hour recall weights were harmonized with examination/interview weights, if applicable.

Response 5: We thank the reviewer for pointing this out. In all analyses, logistic regression models were estimated using the survey design option in STATA (svy: commands), which incorporated stratification and clustering variables (kstrata and PSU) as well as the appropriate weights. Specifically, we applied the wt_tot variable, which is the integrated weight provided by KNHANES to combine the health interview, health examination, and nutrition survey components. For the 2018–2019 combined dataset, we created a composite weight (wt_tot1819) by summing the annual wt_tot values for each cycle, following the KNHANES analytic guidelines. Weighted prevalence estimates are reported in Table 2, and design effects have been considered in all variance estimations.

  1. Reverse causation and temporality.

Comments 6: The cross-sectional design prevents establishing that supplement use preceded frailty. The duration analysis (≥1 year vs <1 year) is suggestive but vulnerable to survivorship and healthy-user biases. Tone down causal language throughout and consider a negative-control exposure (e.g., a supplement unlikely to influence frailty) as a falsification test if available, at least discuss healthy-user bias more explicitly.

Response 6: We fully agree with the reviewer that the cross-sectional design prevents any inference about temporality or causality. A duration analysis (≥1year vs <1year of supplement use) was initially explored but was excluded from the revised manuscript, as the vitamin C–only group was small and further subdivision resulted in unstable estimates that were highly vulnerable to survivorship and healthy-user biases. A formal negative-control exposure was not available in the KNHANES dataset; however, as part of our revised design we included both vitamin C–only users and other supplement users without vitamin C as separate comparator groups. Because these groups share similar health-seeking characteristics but differ in vitamin C intake, this contrast helps to distinguish potential vitamin C–specific associations from those attributable to general supplement use. Nevertheless, we acknowledge that this does not represent a strict negative control, since some other supplements (e.g., vitamin D, calcium, omega-3) may themselves influence frailty outcomes. In line with the reviewer’s suggestion, we have also toned-down causal language throughout the manuscript and more explicitly discussed the potential for reverse causation and healthy-user bias in the Discussion – Limitations section.

  1. Precision and multiplicity.

Comments 7: The highest dose strata (≥2,000 mg/day) likely contain very small numbers with wide CIs; present exact n, weighted %, and instability diagnostics. When presenting multiple dose and duration comparisons and subgroup analyses (age, sex, income), report interaction p-values and address multiplicity (e.g., false discovery rate or emphasize estimation over hypothesis testing).

Response 7: We thank the reviewer for these important comments. In the revised manuscript, we now report the exact number of participants and weighted number for the ≥2,000 mg/day stratum (n = 16, N= 0.04), and we emphasize the wide confidence intervals that reflect statistical instability due to the very small sample size. Because the vitamin C–only group (unweighted number 75) itself was relatively small, we did not conduct further stratified analyses (e.g., by intake duration), as these would have produced unstable estimates with very wide confidence intervals. With respect to multiplicity, we removed several subgroup analyses to avoid overinterpretation.

  1. Dietary intake assessment and measurement error.

Comments 8: A single 24-hour recall is noisy for individual vitamin C intake. You note attempts to match supplement reporting with recall, but serum vitamin C was unavailable. Please expand on expected direction of bias (likely toward the null for diet alone) and consider regression calibration or sensitivity analyses to assess misclassification.  

Response 8: We agree with the reviewer that the use of a single 24-hour recall introduces substantial measurement error in estimating individual dietary vitamin C intake. Serum vitamin C data were not available in the KNHANES 2018–2019 cycles, which limited our ability to conduct regression calibration or validation analyses. Because such measurement error is expected to be largely non-differential with respect to frailty status, the bias would likely attenuate associations toward the null. We have now explicitly acknowledged this limitation and clarified the likely direction of bias in the revised Discussion section (Lines 311–313).

  1. Outcome communication—absolute risks.

Comments 9: Given the relatively low frailty prevalence (5.3% in vitamin C users, 17.2% in non-DS), provide absolute risk differences or adjusted marginal probabilities alongside ORs to aid clinical interpretation.

Response 9: We thank the reviewer for this valuable suggestion. In the revised manuscript, we have supplemented the odds ratio estimates with adjusted predicted probabilities of frailty for each exposure group, derived from fitted logistic regression models using post-estimation predictive margins. We also calculated absolute risk differences (ARDs) with 95% CIs from these margins, and conducted pairwise group comparisons using post-estimation contrast procedures. These results are now presented in the revised Table 3, which facilitates clearer clinical interpretation alongside the relative estimates.

Minor comments

Comments 10: Abstract/Conclusions: Rephrase to avoid causal wording (e.g., “is associated with lower prevalence” rather than “helps manage”). The abstract already uses associative language in places; ensure consistency throughout.

Response 10: We thank the reviewer for this suggestion. We have carefully revised the Abstract and Conclusions to consistenty use associative language and removed any phrasing that could imply causality

Comments 11: Methods clarity: Add a participant-flow figure with explicit counts at each exclusion, including numbers of “other supplement users” removed from the comparator.

Response 11: We have added a flow diagram (Figure 1) that presents explicit counts for all exclusions, including participants who were classified as “other supplement users.”

Comments 12: Reporting: State the Korean RNI (100 mg/day) and Korean UL (2,000 mg/day) in the Methods where dose categories are introduced, with references; the rationale is now scattered across sections. Ensure the 1,000 mg “UL by WHO” phrasing is accurate or corrected.

Response 12: We have clarified the rationale for dose categories in the Methods – Exposure Assessment section. The Korean RNI (100 mg/day) and UL (2,000 mg/day) have been explicitly stated with references, and the previously inaccurate “1,000 mg UL by WHO” phrasing has been corrected (Lines 149–151).

Comments 13: Tables: In Table 2 and dose/duration tables, add survey-weighted means/percentages with 95% CIs. Provide footnotes defining all abbreviations and the exact survey weight used.

Response 13: We revised the tables to include survey-weighted means/percentages with 95% confidence intervals. Footnotes have been expanded to define all abbreviations and specify the exact survey weight applied (Tables 2, 3, 4).

Comments 14: Language/formatting: A handful of phrasing issues (e.g., “adversely linear” → “inverse linear”) and spacing/typography should be polished during revision.

Response 14:  We have carefully reviewed the manuscript and corrected the identified phrasing issues and typographical inconsistencies.

Comments 15: Ethics/data availability: IRB approvals and data availability statements are appropriate; keep accession links stable.

Response 15: We confirm that IRB approvals and data availability statements remain unchanged, and accession links have been verified for stability.

Comments 16: The dataset is nationally representative, and the authors appropriately mention complex survey design. However, key aspects of exposure definition, sample selection, potential healthy-user/selection biases, and interpretation need substantial revision before the paper is reliable for publication. I recommend major revision.

Response 16: We appreciate the reviewer’s overall assessment and constructive feedback. We have substantially revised the manuscript in accordance with these comments, including clearer exposure definitions, transparent reporting of sample selection, and more cautious interpretation of the findings. We hope that the revisions satisfactorily address the concerns and improve the manuscript’s reliability.

Round 2

Reviewer 1 Report

Comments and Suggestions for Authors

No additional comments

Reviewer 3 Report

Comments and Suggestions for Authors

I endorse the publication after English editing and proofreading.